# Evaluation of the Second Premolar’s Bud Position Using Computer Image Analysis and Neural Modelling Methods

**DOI:** 10.3390/ijerph192215240

**Published:** 2022-11-18

**Authors:** Katarzyna Cieślińska, Katarzyna Zaborowicz, Maciej Zaborowicz, Barbara Biedziak

**Affiliations:** 1Department of Orthodontics and Facial Abnormalities, University of Medical Sciences in Poznan, Colegium Maius, Fredry 10, 61-701 Poznan, Poland; 2Department of Biosystems Engineering, Poznan University of Life Sciences, Wojska Polskiego 50, 60-627 Poznan, Poland

**Keywords:** pantomographic radiograph (PR), artificial neural network (ANN), neural modeling, digital radiographs analysis

## Abstract

Panoramic radiograph is a universally used diagnostic method in dentistry for identifying various dental anomalies and assessing developmental stages of the dentition. The second premolar is the tooth with the highest number of developmental abnormalities. The purpose of this study was to generate neural models for assessing the position of the bud of the second premolar tooth based on analysis of tooth–bone indicators of other teeth. The study material consisted of 300 digital pantomographic radiographs of children in their developmental period. The study group consisted of 165 boys and 135 girls. The study included radiographs of patients of Polish nationality, aged 6–10 years, without diagnosed systemic diseases and local disorders. The study resulted in a set of original indicators to accurately assess the development of the second premolar tooth using computer image analysis and neural modelling. Five neural networks were generated, whose test quality was between 68–91%. The network dedicated to all quadrants of the dentition showed the highest test quality at 91%. The training, validation and test subsets were divided in a standard 2:1;1 ratio into 150 training cases, 75 test cases and 75 validation cases.

## 1. Introduction

In the diagnosis of the development of the dentition, clinicians use a pantomographic radiograph. Pantomographic radiography has become a commonly used imaging implement in dental practice and can be a valuable diagnostic tool. Pantomographic radiograph (panoramic) is a layered image of the alveolar process of the maxilla and mandible created by synchronous rotation of the X-ray tube and radiation detector around the patient’s head. This examination allows us to visualize the curved structures of the skull [1] and simultaneously shows all the teeth, mandibular bones, maxilla’s part, a large part of the maxillary sinuses, hard palate and both temporomandibular joints. The image allows us to assess the patient’s development and the potential need for specialized treatment in the case of developmental defects [2,3]. Pantomographic radiograph (PR) is a common diagnostic method used in dentistry to identify hard tissues of the oral cavity and skeletal structures. The diagnostic value of such radiographs is very high; however, analysis of the images is demanding. Dentists are very well acquainted with the image of bones and teeth, but sometimes it is challenging and time-consuming to interpret the image and identify all the structures visible in such a radiograph [4]. The resolution is not as accurate as intraoral radiographs, but nevertheless allows detection of caries, periapical lesions, impacted teeth, bone fractures, neoplastic lesions, and allows assessment of the dental status and developmental stages of the dentition. Pantomographic radiograph also allows assessment of the presence of dental anomalies such as abnormalities in the number of teeth, abnormalities in the shape of teeth, abnormalities in the structure of the hard tissues of the tooth, abnormalities in the topography and timing of eruption [5].

The second premolar is the tooth in which the most developmental disorders occur such as agenesis, retention, distal tilt causing damage of first molar. The formation of the second premolars’ buds is often delayed [6,7]. It is justifiable to prepare a device that efficiently determines the position of the second premolar’s buds and can help plan appropriate orthodontic treatment. 

A metric analysis of pantomographic radiographs designed to predict the development of the second premolar was attempted. The paper presents a method for evaluating the position of the second premolar’s bud based on the author’s parameters. This work is a continuation of earlier studies describing the analysis of radiographs to assess the pattern of eruption of canines and premolars [8].

## 2. The Aim

The purpose of the study was to generate neural models for assessing the position of the bud of the second premolar tooth based on analysis of tooth–bone indicators of other teeth.

### 2.1. Material

The research material consisted of digital pantomographic radiographs of children in the developmental period. The age of the patients was 6–10 years. A total of 300 radiographs were analyzed. The study group included 165 boys and 135 girls.

The radiographs were obtained from the database of the University Centre of Dentistry and Specialist Medicine in Poznań, Poland. The Bioethics Committee of the Poznan University of Medical Sciences concluded that the research did not have the characteristics of a medical experiment and therefore approved the relevant work.

The radiographs used in the study were of patients of Polish nationality, aged 6–10 years (72–120 months), with no known systemic diseases, lesions or defects in the craniofacial region, no dental malformations and no diseases of the hard tissues of the teeth and pulp.

The pantomographic radiographs used in the study were taken with a German Durr Dental camera (VistaPano S Ceph), which is equipped as standard with an X-ray head with a 0.5 mm focus, a digital sensor and a Cls-CMOS matrix, and additionally enables the taking of images with S-PAN technology, giving the images better sharpness and saturation of mineralized structures. Specialized software DBSWIN (Durr Dental) is designed to read digital images recorded by the X-ray camera in DICOM 3.0 format. The application runs on MS Windows. The software allows one to create, import and export data and image databases. It reads 16-bit images of 65,536 shades of gray and is dedicated to the presentation of medical radiographs [9].

Measurements of tooth and bone parameters were made with the open-source program ImageJ 1.8v. (LOCI University of Wisconsin) [10], and then the measurements determined on the pantomographic radiographs were entered into the database.

The collected study material was reviewed and radiographs with developmental abnormalities were excluded. Patient data with age determined in months were entered into an MS Excel spreadsheet (Microsoft Excel 15.0.4420.1017) [11].

### 2.2. Methods

Relying on the established points, distances were determined which allowed the subsequent development of the tooth and bone indicators (teeth were marked according to FDI’s international system). Subsequently, the indicators were subjected to a neural modeling process.
*X*-axis—Left vertical edge of the pantomographic radiograph.U—Width of the mouthpiece of the X-ray equipment.KN-BZ—Distance of the anterior nasal spike from the mandibular margin.

Graphic representation of the distances is in the Figure 1.

S11-W11 is the distance between point S11 (on the incisal margin of tooth 11) from point W11 (the point on the alveolar margin in the line drawn through the point on top of the alveolus perpendicular to the X line).

S41-W41 is the distance between point S41 (on the incisal margin of tooth 41) from point W41 (the point on the alveolar margin in the line drawn through the point on top of the alveolar process perpendicular to line X).

Graphic representation of this distances is in the Figure 2.

S12-W12 is the distance between point S12 (on the incisal margin of tooth 12) from point W12 (the point on the alveolar margin in the line drawn through the point on top of the alveolus perpendicular to the X line).

S42-W42 is the distance between point S42 (on the incisal margin of tooth 42) from point W42 (the point on the alveolar margin in the line drawn through the point on top of the alveolar process perpendicular to line X).

Graphic representation of this distances is in the Figure 3.

S13-W13 is the distance between point S13 (on the incisal margin of tooth 13) from point W13 (the point on the alveolar margin in the line drawn through the point on top of the alveolus perpendicular to the X line).

S43-W43 is the distance between point S43 (on the incisal margin of tooth 43) from point W43 (the point on the alveolar margin in the line drawn through the point on top of the alveolar process perpendicular to line X).

Graphic representation of the distances is in the Figure 4.

S14-W14 is the distance between point S14 (on the incisal margin of tooth 14) from point W14 (the point on the alveolar margin in the line drawn through the point on top of the alveolus perpendicular to the X line).

S44-W44 is the distance between point S44 (on the incisal margin of tooth 44) from point W44 (the point on the alveolar margin in the line drawn through the point on top of the alveolar process perpendicular to line X).

Graphic representation of the distances is in the Figure 5.

S15-W15 is the distance between point S15 (on the incisal margin of tooth 15) from point W15 (the point on the alveolar margin in the line drawn through the point on top of the alveolus perpendicular to the X line).

Measurements of each unitary teeth in the other quadrants were made in a similar manner.

Based on the designated distances, mathematical proportions were developed as tooth and bone indicators. Indicators in the form of mathematical proportions are presented in Table 1:

The next indicator is a formula that determines the eruption of the canine, first and second premolars in each quadrant. An erupted tooth was referred to as 0 and an unerupted tooth as 1. For each quadrant, the arrangement of teeth was determined according to this principle: A: 0-0-0; B:0-0-1; C: 0-1-1; D:1-0-0; E: 0-1-1; F: 1-0-0; G: 1-1-0; H: 1-1-1.

The last indicator determined the position of the buds relative to the alveolar margin of the maxilla or the alveolar part of the mandible based on a comparison of the lengths of the designated distances, in the order from the bud located closest to the bony margin to the one located furthest from it. For each quadrant, the arrangement of the buds of the first and second premolar and the canine were determined: A: 4-5-3; B: 4-3-5; C: 5-4-3; D: 5-3-4; E: 3-4-5; F: 3-5-4.

On the basis of the conducted tooth and bone measurements, learning sets were created. Appendix A presents the matrix of variables included in the learning sets for quadrants I-IV. The first 10 variables, GENDER, AGE, AV, CQ, CS, CU, CW, CY, DA, DC and DE were identical in each of the 4 sets, while the next 5 variables were specific to each quadrant. The last variable in each set was the output of the network. An additional learning set was created as a combination of sets for quadrants I-IV. It contained 31 variables, of which 27 were input variables. The variables DI, DM, DU and DQ constituted the output of the network.

Subsequently, the function Automatic Network Designer was used and the basic parameters for creating ANN models were defined. In the process of neural modeling, RBF (radial basis function) networks and MLP (multilayer perceptron) three- and four-layer perceptrons were tested. The 10 best networks were retained. The criteria for retaining networks was the balance between error and network diversity. If the result set was filled with models with similar characteristics - the set was to be expanded. The number of networks tested was 20 for every type. STATISTICA (TIBCO Software Inc., Palo Alto, CA, USA) [12] was used to simultaneously determine the set of independent variables and perform sensitivity analysis of the variables. For perceptron-type models, a linear activation function was used; in terms of network complexity, for MLP-type models it was from 1 to 13 neurons in the hidden layer, while for RBF-type networks it was from 1 to 75 neurons. The training, validation and test subsets were divided in a standard 2:1:1 ratio into 150 training cases, 75 validation cases and 75 test cases. The variables for each subset were selected randomly by the STATISTICA simulator algorithm.

### 2.3. Analysis of the Collected Data

During the process of generating Artificial Neural Networks models, five learning sets were used, prepared from previously conducted tooth and bone measurements: LS 2021,04,25 Q-I; LS 2021,04,25 Q-II; LS 2021,04,25 Q-III; LS 2021,04,25 Q-IV; and ZU 2021,04,25 4Q. The LS 2021,04,25 4Q set was a combination of the LS 2021,04,25 I–IV sets.

LS means learning set, 2021,04,25 indicates the date of its creation, Q is the abbreviation for quadrant, I-IV is the number of the quadrant and 4Q is all quadrants in one file.

There were 300 teaching cases (clinical cases) in each dataset, which were patients aged 6–10 years (72–120 months). The group included 135 girls and 165 boys.

Learning sets are CSV files, containing the following variables:LS 2021,04,25 Q-I
-NETWORK INPUT: GEDER, AGE, AV, CQ, CS, CU, CW, CY, DA, DC, DE, DG, DW, ED, EL-NETWORK OUTPUT: DILS 2021,04,25 Q-II
-NETWORK INPUT: GENDER, AGE, AV, CQ, CS, CU, CW, CY, DA, DC, DE, DK, DY, EE, EP-NETWORK OUTPUT: DMLS 2021,04,25 Q-III
-NETWORK INPUT: GENDER, AGE, AV, CQ, CS, CU, CW, CY, DA, DC, DE, DS, EC, EG, EX-NETWORK OUTPUT: DULS 2021,04,25 Q-IV
-NETWORK INPUT: GENDER, AGE, AV, CQ, CS, CU, CW, CY, DA, DC, DE, DO, EA, EF, ET-NETWORK OUTPUT: DQLS 2021,04,25 4Q
-NETWORK INPUT: GENDER, AGE, AV, CQ, CS, CU, CW, CY, DA, DC, DE, DG, DK, DO, DS, DW, DY, EA, EC, ED, EE, EF, EG, EL, EP, ET, EX-NETWORK OUTPUT: DM, DI, DU, DQ

### 2.4. Neural Modeling

In the neural modelling process, the goal was to generate a network with the highest possible score for the test set and the lowest possible error for the test set. The learning set was used to generate and train the model. The validation set was used to tune the weights of the network while the test set was not involved in the process of generating the network. The model was checked against the cases included in the test set. By network error, we mean the RMSE error—that is, the root of the mean square error.
$$\mathrm{RMSE}=\sqrt{\frac{1}{N}\sum_{i=1}^{N}{\left(t_{i}-y_{i}\right)}^{2}}$$

#### 2.4.1. LS2021,04,25 Q-I

For the set LS 2021,04,25 Q-I, 10 networks were generated (Appendix A) The optimal model for determining the DI parameter was the RBF 14:14-5-1:1 network. It was noticed that the higher quality indicators for the test set were characterized by the MLP 1:1-1-1:1 network, but the RMSE error for this set was at the level of 22%. The RBF 14:14-5-1-1:1 network had a quality for the test set of 89% and an error for the test set of 12%. The network needed 14 variables for its operation. A summary of the variables and a sensitivity analysis of the network are provided in Appendix A. A diagram of the network is shown in Appendix A.

#### 2.4.2. LS 2021,04,25 Q-II

For the set LS 2021,04,25 Q- II, 10 networks were generated (Appendix A). The optimal model for determining the DM parameter was the RBF 12:12-5-1:1 network. The RBF 12:12-5-1:1 network had a quality for the test set of 89% and an error for the test set of 14%. The network needed 12 variables for its operation. A summary of the variables and sensitivity analysis of the network are provided in Appendix A. The network diagram is shown in Appendix A.

#### 2.4.3. LS 2021,04,25 Q-III

For the set LS 2021,04,25 Q-III, 10 networks were generated (Appendix A). The optimal model for determining the DU parameter was the RBF 13:13-10-1:1 network. The RBF 13:13-10-1:1 network had a quality for the test set of 69% and an error for the test set of 7%. The network needed 13 variables for its operation. A summary of the variables and a sensitivity analysis of the network are provided in Appendix A. A diagram of the network is shown in Appendix A.

#### 2.4.4. LS 2021,04,25 Q-IV

For the set LS 2021,04,25 Q-III, 10 networks were generated (Appendix A). The optimal model for determining the DQ parameter was the RBF 10:10-10-1:1 network. The RBF 10:10-10-1:1 network had a quality for the test set of 73% and an error for the test set of 8%. The network needed 10 variables for its operation. A summary of the variables and sensitivity analysis of the network are provided in Appendix A. The network diagram is shown in Appendix A.

#### 2.4.5. LS 2021,04,25 I-IV

For the LS 2021,04,25 I-IV sets, the modeling process was identical. Continuous variables were defined as AGE, AV, CQ, CS, CU, CW, CY, DA, DC and DE, and there were another 6 quadrant-specific variables, as well as a categorized variable: GENDER.

#### 2.4.6. Summary of the Models for Each Quadrant

In order to determine the distance of the bud of the second premolar tooth from the edge of the alveolar margin, it was necessary to use new, original tooth–bone indicators and Artificial Neural Network models dedicated to each quadrant of the dentition. A summary of the generated models and their quality indicators is provided in Appendix A.

The highest network characteristics were achieved for models determining tooth and bone parameters in the first and second quadrants. The quality of the network for the test set was more than 88%, while the RMSE error for the test set did not exceed 14%. Worse model performance was obtained for quadrants III and IV; the quality of the network for the test set was about 70%, while the error was about 7%.

All the models generated were RBF networks—networks with radial basis functions. The use of this type of network demonstrates the nonlinear nature of the issue.

Appendix A presents the variables used in the neural modeling process for each quadrant and model.

The variables that were repeated for each quadrant, and were used in the neon modeling process, were: GENDER, AV, CS and CW. Quarter-specific variables were also used for each quadrant: for the first quadrant they were DG, DW, ED and EL; for the second quadrant they were DK, DY, EE and EP; for the third quadrant they were DS, EC, EG and EX; and for the fourth quadrant they were DO, ES, EF and ET.

#### 2.4.7. LS 2021,04,25 4Q

Continuous variables were defined for the LS 2021,04,25 4Q set as: AGE; AV; CQ; CS; CU; CW; CY; DA; DC; DE; DG; DK; DO; DS.; DW; DY; EA; EC; ED; EE; EF; EG; EL; EP; ET; EX; DI; DM; DU; and DQ. There was also a categorized variable: GENDER.

For the LS 2021,04,25 4Q set, 10 networks were generated, the characteristics of which are presented in Appendix A.

The two most effective models generated were MLP 26:26-18-18-4:4 and RBF 19:19-15-4:4. The MLP network was a multilayer perceptron with a high quality for the test set, at 99%, and an RMSE error for this set of nearly 26%. In contrast, the RBF 19:19-15-4:4 type model had a quality for the test set of 91% and an RMSE error for this set of 8%. Due to the lower value of the RMSE error, the suggested model is RBF. The RBF 19:19-15-4:4 network has a simpler topology. It needs only 19 variables (out of 27 input variables) for its operation. The network has 4 outputs defining the parameters DI, DM, DU and DQ specific to each quadrant.

The sensitivity analysis of the network made it possible to determine which of the variables used were the most significant and which were the least significant. A relevant summary is provided in Appendix A presents a schematic of the generated network.

## 3. Summary of Models

A summary of the generated models can be found in Appendix A. The criteria for selecting the network were the highest possible quality for the test set and the lowest possible error for this set (the quality of the network is understood as the difference between the real (empiric) value of the network and the output network value).

The RBF 19:19-15-4:4 network with 19 inputs and 4 outputs had a quality of 91% and an RMSE error of 8%. Other networks dedicated to each quadrant were characterized by worse quality parameters. It should be noted that the models for quadrants I and II were characterized by higher quality indicators than those for quadrants III and IV. The difference was as much as 20 percentage points for set quality. However, the models for quarters III and IV were characterized by lower RMSE error values.

The matrix of variables used to generate the network and the number of matching variables for each model are shown in Appendix A.

The most important variables common to all the quadrants (occurring at least three out of five times) were: GENDER; AGE; AV; CQ; CS; CW; DC; and DE. Variables that characterized the other quadrants (occurring two times) were: DW; ED; DK; DY; EE; EP; DS; EC; EG; EX; DO; EA; EF; and ET.

## 4. Discussion

Accurate analysis of a pantomographic radiograph requires time and expertise. As a support in the diagnostic evaluation of PR, computer-assisted systems have been developed to facilitate and improve image analysis and thus treatment planning [13].

Researchers Tuzoff et al. analyzed pantomographic radiographs using a convolutional neural network, which performed automatic analysis of panoramic radiographs for tooth detection and numbering. A data set of 1352 randomly selected adult pantomographic radiographs was used to train the artificial neural network. Each radiograph was processed accordingly to determine the borders of each tooth. The tooth numbering module classified the detected tooth images according to FDI designations. The performance of the proposed computer-assisted diagnostic solution was comparable to that of experts, and the performance indices were 99% sensitivity and 99% accuracy in tooth detection, and 98% sensitivity and 99% accuracy in tooth numbering. According to the authors, this method has potential for practical application and further evaluation for automated analysis of dental radiographs. Computerized tooth detection and numbering simplifies the process of completing digital dental diagrams [14].

A similar study was conducted by Korean researchers, who presented a technique combining neural network, multibox single shot detector and heuristic methods for detecting teeth and implants from pantomographic radiographs. The authors of the paper presented an algorithm that detects dental objects in a panoramic radiograph and determines the tooth number based on its shape and location. The study showed high sensitivity of the neural network in detecting crowns and implants (96%) and sensitivity, specificity and accuracy in tooth numbering of 84%, 75% and 84%, respectively. The work was aimed at simplifying the interpretation of the images and helping the patient understand the diagnostic information [15]. An artificial neural network was used to detect dental abnormalities in pantomographic radiographs. Researchers from Korea, En-Gyu Ha et al., used the capabilities of the artificial neural network to detect the presence of a supernumerary tooth in the maxilla—mesiodens. The spliced neural network used (YOLO—you only look once) demonstrated high efficiency in detecting mesiodens on images of three groups of dentition: deciduous, mixed and permanent. Performance of about 96%, 97% and 93% for each group of dentition, respectively, demonstrates the potential for clinical application of artificial intelligence as a tool to assist clinicians [16]. Deep convolutional neural networks are gaining increased attention in the field of medical imaging. A deep learning tool for image detection, YOLO, is characterized by its simple data processing network, which can both detect and classify an object at the same time. A study presented by researchers in Korea showed that a real-time CNN YOLO detector trained on a limited number of labeled panoramic images showed diagnostic performance at least similar to that of experienced dentists in detecting odontogenic cysts and tumors on a pantomographic radiograph, and its precision was 70% [17].

Not only can radiographs be analyzed, but the detector can also be used in medicine to evaluate other types of tests such as EEG waveform. In their study using the convolutional network YOLO v3, Natheer Khasawneh et al. analyzed EEG waveform to detect K-complex. Detection of K-complex is important for diagnosing neurophysiologic and cognitive disorders and sleep studies, but it is difficult and time-consuming. Based on multiple convolutional neural network (CNN) you only look once v3 (YOLOv3), the detector demonstrated high accuracy in automatic K-complex detection (precision 99%). This means that such a tool will be helpful in the work of practitioners in efficient EEG verification [18].

Another study by Dian Pratiwi demonstrated the ability of an artificial neural network to detect abnormal tooth structure in pantomographic radiographs. The artificial neural network trained by the researchers was shown to be a promising tool for demonstrating defects in tooth structure. In cross-validation tests, the accuracy was 80–86%. According to the author, the study needs further elaboration, but it gives an idea of the possibilities of the ANN as a support in the work of the doctor [19]. An attempt was also made to use an artificial neural network in the analysis of radiographs to determine the gender of patients. Differences in the bone structure of men and women were the basis for training the neural network. The accuracy of the ANN’s gender estimation was 90–96%, despite the fact that this result applies only to images of patients over 20 years of age, in which there are clear differences in structure visible on radiographs [20].

Analysis of pantomographic radiographs using deep neural modeling was conducted by Zaborowicz et al. to estimate metric age. The researchers determined 21 tooth and bone parameters that allowed a fast and precise assessment of a patient’s metric age. Previously used methods of assessing metric age based on tables or charts were time-consuming, and the use of new computer techniques is proving to be a suitable tool to improve the doctor’s work [21].

## 5. Conclusions

As a result of the conducted research, a set of 23 tooth–bone indicators was identified and presented, with which it is possible to identify the position of a second premolar’s bud from digital panoramic radiographs, using a neural modeling method.

The models produced are RBF networks—networks with radial basis functions. The use of this type of network demonstrates the nonlinear nature of the problem. Five neural networks were generated, whose test quality was between 68–91%. The RBF 19:19-15-4:4 network dedicated to all quadrants of the dentition presented the highest quality for the test set at 91% and an RMSE error of 8%. RBF network models 14:14-5-1:1 for quadrant I and 12:12-5-1:1 for quadrant II presented quality for the set at 89% and RMSE error of 12% and 14%, respectively. In contrast, the 13:13-10-1:1 and 10:10-10-1:1 network models for quarters III and IV indicated lower quality parameters, at 69% and 72%, respectively. The lowest RMSE error of 7% was observed for quarter III of the dentition.

The study shows that the neural modeling method is a suitable tool for determining the position of the buds of the second premolar. The developed methodology can be used as an algorithm for implementation in a computer application that will automatically determine the position of buds on panoramic radiographs of children and adolescents aged 6 to 10 years. It is a promising tool to support clinicians in their work.

## Figures and Tables

**Figure 1 ijerph-19-15240-f001:**
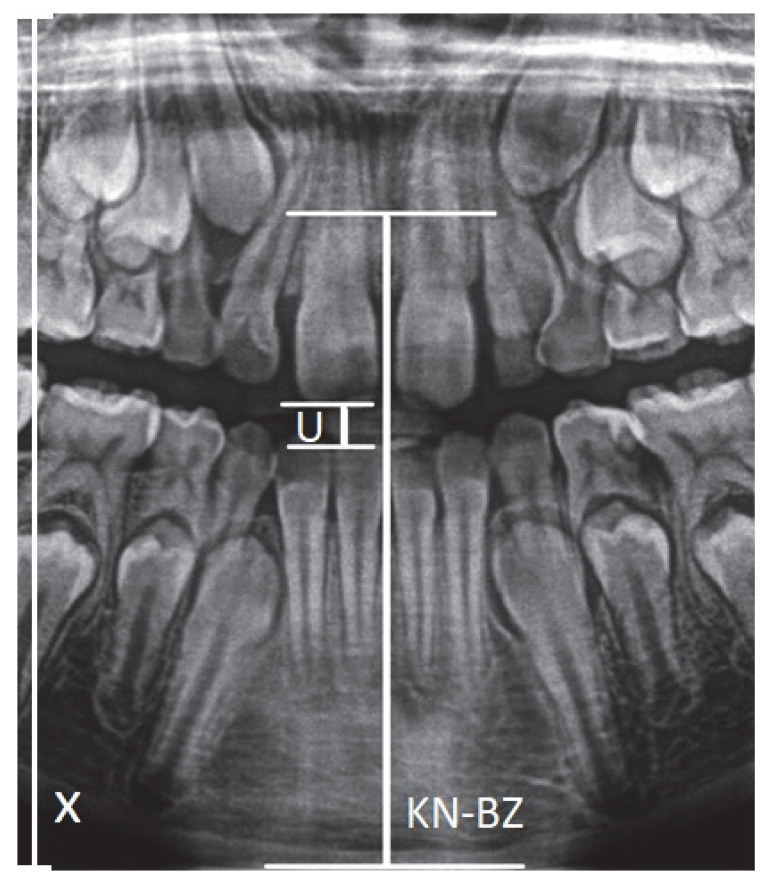
Graphic representation of the KN-BZ and U distances (KN-BZ—Distance of the anterior nasal spike-KN from the mandibular margin-BZ; U—Width of the mouthpiece of the X-ray equipment).

**Figure 2 ijerph-19-15240-f002:**
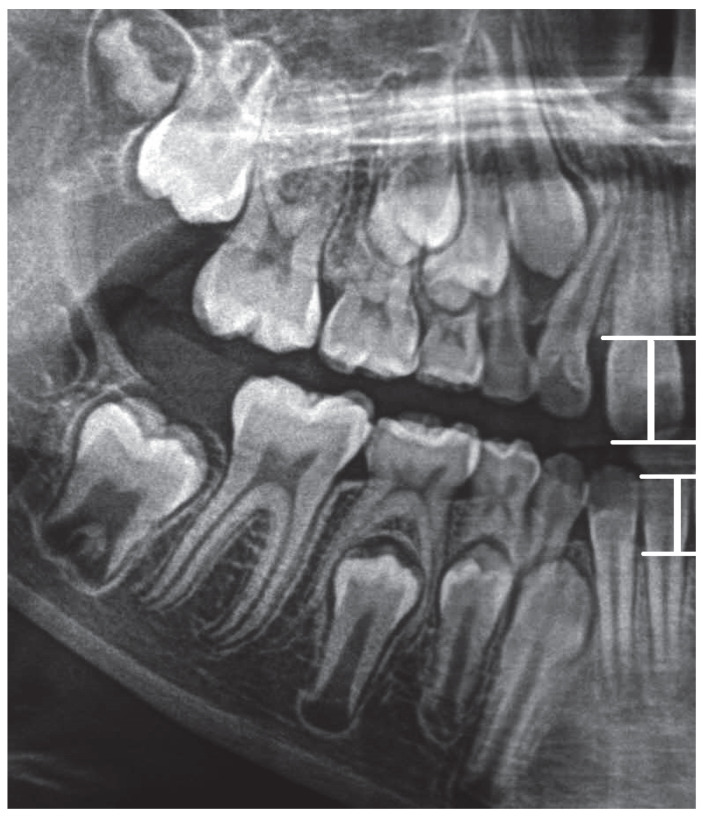
Graphic representation of the S11-W11 and S 41-W41 distances.

**Figure 3 ijerph-19-15240-f003:**
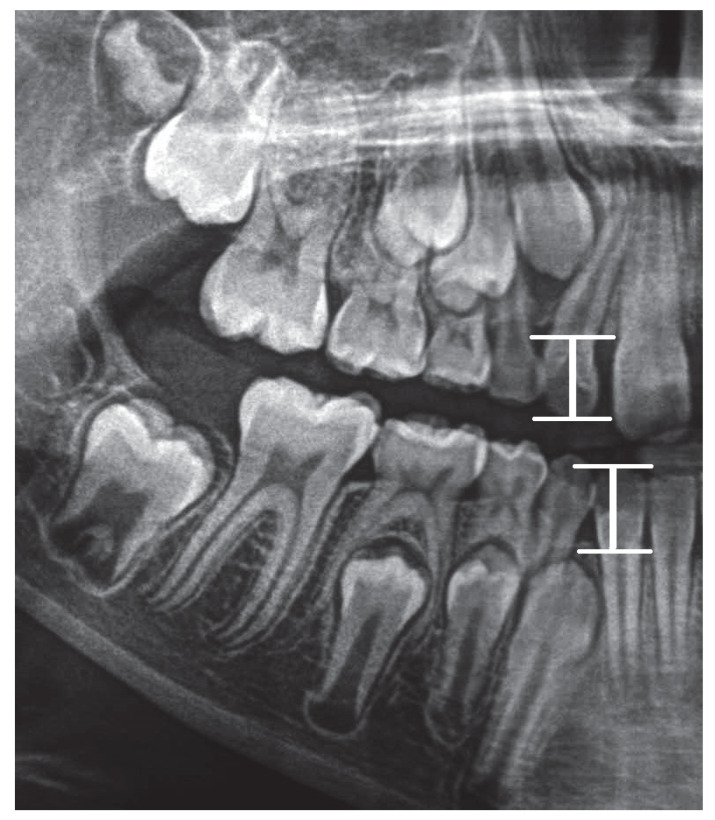
Graphic representation of the S12-W12 and S 42-W42 distances.

**Figure 4 ijerph-19-15240-f004:**
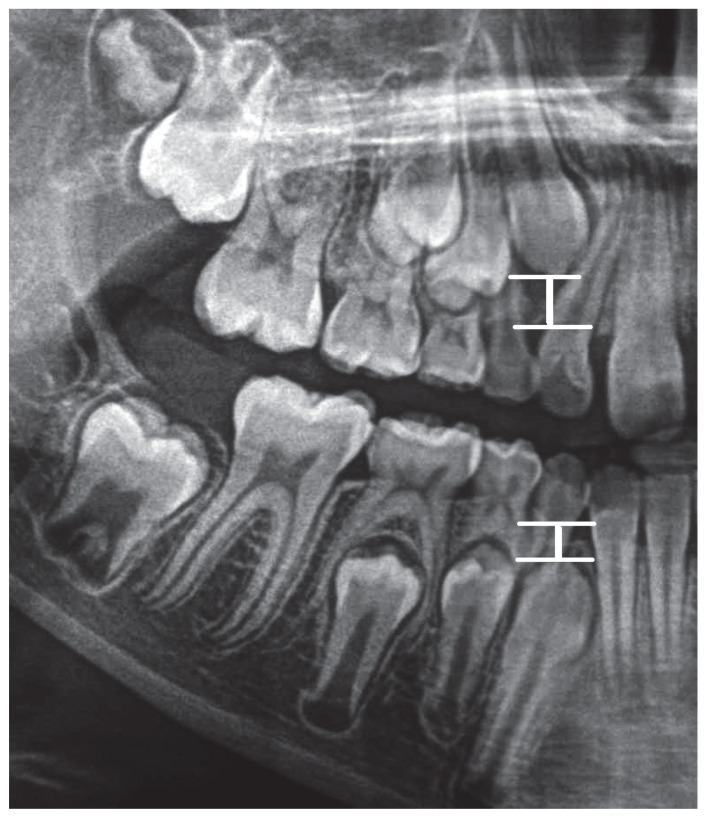
Graphic representation of the S13-W13 and S 43-W43 distances.

**Figure 5 ijerph-19-15240-f005:**
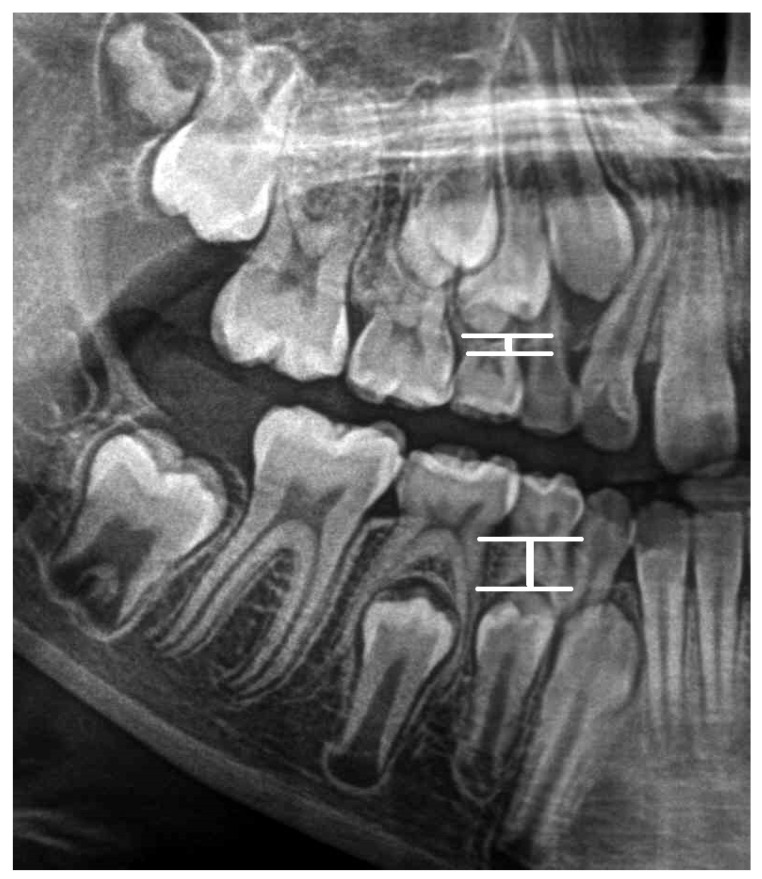
Graphic representation of the S14-W14 and S 44-W44 distances.

**Table 1 ijerph-19-15240-t001:** Indicators in the form of mathematical proportions.

No.	Description of Ratio	Ratio
1	AV—the ratio between the distance |KN-BZ| and U (the width of the interdental handle-mouth of the X-ray camera)	$\mathrm{AV}\;=\frac{\left(\mathrm{KN}-\mathrm{BZ}\right)}{U}$
2	CQ—the ratio between S11-W11 and AV distance	$\mathrm{CQ}=\frac{\left(S11-W11\right)}{\mathrm{AV}}$
3	CS—the ratio between S12-W12 and AV distance	$\mathrm{CS}=\frac{\left(S12-W12\right)}{\mathrm{AV}}$
4	CU—the ratio between S21-W21 and AV distance	$\mathrm{CU}=\frac{\left(S21-W21\right)}{\mathrm{AV}}$
5	CW—the ratio between S22-W22 and AV distance	$\mathrm{CW}=\frac{\left(S22-W22\right)}{\mathrm{AV}}$
6	CY—the ratio between S31-W31 and AV distance	$\mathrm{CY}=\frac{\left(S31-W31\right)}{\mathrm{AV}}$
7	DA—the ratio between S41-W41 and AV distance	$\mathrm{DA}=\frac{\left(S41-W41\right)}{\mathrm{AV}}$
8	DC—the ratio between S32-W32 and AV distance	$\mathrm{DC}=\frac{\left(S32-W32\right)}{\mathrm{AV}}$
9	DE—the ratio between S42-W42 and AV distance	$\mathrm{DE}=\frac{\left(S42-W42\right)}{\mathrm{AV}}$
10	DG—the ratio between S14-W14 and AV distance	$\mathrm{DG}=\frac{\left(S14-W14\right)}{\mathrm{AV}}$
11	DK—the ratio between S24-W24 and AV distance	$\mathrm{DK}=\frac{\left(S24-W24\right)}{\mathrm{AV}}$
12	DS—the ratio between S34-W34 and AV distance	$\mathrm{DS}=\frac{\left(S34-W34\right)}{\mathrm{AV}}$
13	DO—the ratio between S44-W44 and AV distance	$\mathrm{DO}=\frac{\left(S44-W44\right)}{\mathrm{AV}}$
14	DW—the ratio between S13-W13 and AV distance	$\mathrm{DW}=\frac{\left(S13-W13\right)}{\mathrm{AV}}$
15	DY—the ratio between S23-W23 and AV distance	$\mathrm{DY}=\frac{\left(S23-W23\right)}{\mathrm{AV}}$
16	EC—the ratio between S33-W33 and AV distance	$\mathrm{EC}=\frac{\left(S33-W33\right)}{\mathrm{AV}}$
17	EA—the ratio between S43-W43 and AV distance	$\mathrm{EA}=\frac{\left(S43-W43\right)}{\mathrm{AV}}$
18	DI—the ratio between S15-W15 and AV distance	$\mathrm{DI}=\frac{\left(S15-W15\right)}{\mathrm{AV}}$
19	DM—the ratio between S25-W25 and AV distance	$\mathrm{DM}=\frac{\left(S25-W25\right)}{\mathrm{AV}}$
20	DU—the ratio between S35-W35 and AV distance	$\mathrm{DU}=\frac{\left(S35-W35\right)}{\mathrm{AV}}$
21	DQ—the ratio between S45-W45 and AV distance	$\mathrm{DQ}=\frac{\left(S45-W45\right)}{\mathrm{AV}}$

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
