# Peer review of "Evaluation of the Second Premolar’s Bud Position Using Computer Image Analysis and Neural Modelling Methods"

_ijerph, 2022, doi:10.3390/ijerph192215240_

Round 1

Reviewer 1 Report (Previous Reviewer 1)

The manuscript has been improved after revision.

The figure caption language should be improved with a self-explained description. For example in Figure 5, what are the whisker bars indicating, and the values? Including the point label on the figures would be easy for readers. The word "sections" should be 'distances' or 'segments' in my opinion. Every figure and table should be referenced in the main text.

Repetative descripition of S11-W11, S21-W21, S31-W31, S41-W41 are not necessary.

In table 2, many abbreviations such as AV, CQ, DC, W14, etc. are missing.

The whole manuscript should be proofread to correct inconsistencies, typography, and unclear statements. Examples: "L108 S31-W41 is the distance" "L203 20,210,425 - means the date of its creation" .'L187 he last indicator determines the'

Major result data should be presented and summarized in the main text, not supplementary. 

The reference list is not in a proper format. Reference #8 cannot be found. Reference list #9 and 10 are not informative. 

The teaching cases of 150 seems a very low number. 

Author Response

Thank you very much for your time and valuable comments on the text.

The figure caption language should be improved with a self-explained description. For example in Figure 5, what are the whisker bars indicating, and the values? Including the point label on the figures would be easy for readers. The word "sections" should be 'distances' or 'segments' in my opinion. Every figure and table should be referenced in the main text.

The word ‘’sections” has been changed and replaced with the word  ‘distances’

References to tables and figures have been marked in the text with the yellow colour.

Repetative descripition of S11-W11, S21-W21, S31-W31, S41-W41 are not necessary.

Repeated descriptions have been shortened and for clarity of the text we have left the description relating to the photo.

In table 2, many abbreviations such as AV, CQ, DC, W14, etc. are missing.

The abbreviations AV,CQ, DC are described in Table 1. As suggested, they were also placed in Table 2 so that all abbreviations are in one place at the end of the text.

The whole manuscript should be proofread to correct inconsistencies, typography, and unclear statements. Examples: "L108 S31-W41 is the distance" "L203 20,210,425 - means the date of its creation" .'L187 he last indicator determines the'

Thank you for your insightful comments. While editing, autocorrect changed the text which was unfortunate overlooked. All errors have been corrected, which is indicated in the text.

Major result data should be presented and summarized in the main text, not supplementary. 

In the original version of the article, all outcome data were presented in the main text. At the suggestion of the reviewers, we created supplementary material into which the figures and tables suggested by the reviewers were transferred. However, if there is such a recommendation we will place all or part of the tables in the main text.

The reference list is not in a proper format. Reference #8 cannot be found. Reference list #9 and 10 are not informative. 

The reference list has been revised and corrected properly. Reference list #9 and 10 are not informative, but I have taken the liberty of leaving  them as I think it may be useful for other authors.

The teaching cases of 150 seems a very low number. 

The study used 300 pantomographic images, of which 50% were used for training and 25% each for validation and testing, respectively. The database is not extensive but this is preliminary research and the plan is to expand the study to include more cases.

Thak you very much.

Reviewer 2 Report (New Reviewer)

- The authors should have checked their manuscript before submission. The highlighting of certain text and the red coloring of some words is weird, or maybe the paper is to show revisions that were made! 

- The authors need to check the tempalate for the proper journal formatting.

- The English language need extensive editing from a computer science perspective. For example a neural network is trained not taught. Training is the correct technical term. 

- Language mistakes throughout the paper, for example, see line 30 " a commonly used imaging implement...", and line 176 "mathematical proportions " line 187 " he last indicator"

- How are the parameters being measured? manually from the images?

- The neural modeling part need extensive revisio as it does not display any computer science properties. This is evident further by the simple mistakes in reported AI methods, line 471 " (YLOO - You Look Only Once)" --> YOLO: you only look once. 

- The performance metrics need to be defined. 

Author Response

Thank you very much for your time and valuable comments on the text.

- The authors should have checked their manuscript before submission. The highlighting of certain text and the red coloring of some words is weird, or maybe the paper is to show revisions that were made! 

- The authors need to check the tempalate for the proper journal formatting.

The highlighted text and red markings are due to the fact that we were asked to show the corrections made according to the recommendations of previous reviewers. The text was checked and formatted according to the publisher's guidelines

- The English language need extensive editing from a computer science perspective. For example a neural network is trained not taught. Training is the correct technical term. 

- Language mistakes throughout the paper, for example, see line 30 " a commonly used imaging implement...", and line 176 "mathematical proportions " line 187 " he last indicator"

Thank you very much for your insightful analysis of the text. All errors have been corrected and changes highlighted in yellow

- How are the parameters being measured? manually from the images?

The measurements were made using ImageJ software. The exact process of performing the measurements is described in the article Eruption pattern of permanent canines and premolars in Polish children. Int. J. Environ. Res. Public Health 2022 in the section describing Material and methods and includes a description of the statistical analysis.

- The neural modeling part need extensive revisio as it does not display any computer science properties. This is evident further by the simple mistakes in reported AI methods, line 471 " (YLOO - You Look Only Once)" --> YOLO: you only look once. 

- The performance metrics need to be defined. 

Thank you very much for your remarks,  relevant changes have been included in the text. The performance metrics have been added.

Quality - The quality of the network is understood as the difference between the real (empiric) value of the network and the output network value.

RMSE - Root Mean Square Error

Round 2

Reviewer 2 Report (New Reviewer)

The author responded to most of my comments. I would suggest including some references about Yolo and its use in the medical literature. See

Detection of K-complexes in EEG signals using deep transfer learning and YOLOv3. Cluster Comput (2022). https://doi.org/10.1007/s10586-022-03802-0

For their future work, I strongly recommend the researchers to include Artificial intelligence specialists in their work, which will greatly improve its quality and introduce new ideas.

Author Response

Thank you very much for your valuable comments. Indeed, the topic of YOLO is extremely interesting and captivating and the cooperation of medical practitioners and artificial intelligence specialists is inevitable in the near future.

Best Regards, Katarzyna Cieślińska

This manuscript is a resubmission of an earlier submission. The following is a list of the peer review reports and author responses from that submission.

Round 1

Reviewer 1 Report

The tables can be combined and the text can be shortened. Please leave the less important supporting table or screenshot in the supplementary.  The research topic is of interest and the workflow seems good to me. 

In the abstract, please indicate the number of cases used for training and the number of cases for testing, and if they are the same pool of cases. 

Author Response

Dear Rewiever

Thank you very much for your time and all your comments regarding the text.

As suggested to make the work more readable, the tables have been placed in the supplementary materials and the screenshots removed.

The relevant information in the abstract regarding the number of cases used in the work was inserted as suggested.

Thank you very much.

Reviewer 2 Report

Thank you for the opportunity to review this interesting paper. While I must credit the hard work done by the authors, the paper in it current form is very difficult to comprehend and read for the average reviewer. It is far too verbose and many data of the parameters defined can easily be better presented but unfortunately it is not done so. Additionally very limited discussion is done on the novelty of the paper rather it is drawing a tangential focus to other research work which again in my humble opinion require extensive editing and a rewrite

Author Response

Dear Reviewer,

Thank you very much for your time and all your comments regarding the text.

I understand that the text may seem difficult and therefore has been analyzed and corrected,the purpose of the study was made more precise and the tables and figures have been moved to supplementary materials. I think it will be more readable in its current form.

Once again, thank you very much for your comments.

Reviewer 3 Report

Very difficult to read- needs extensive English editing. This is far too technical for the general readership, and the methods and results section is too detailed.  I would suggest you should move the majority of the methods and results to supplementary material, focus on the main neural networks in the methods, and provide more detailed annotated images to show which measurements were performed. 

“The Bioethics Committee of the Poznan 69 University of Medical Sciences concluded that the research did not have the characteristics 70 of a medical experiment and therefore approved the relevant work. “

There was a waiver of consent? Unclear

“Patient data with age determined in months were entered into an 89 MS Excel spreadsheet (Microsoft Excel 15.0.4420.1017). This is a tool from the MS Office 90 family of programs that allows the collation and organization of data obtained from the 91 analysis of pantomographic radiographs [11]. “

I personally don’t think this detail is necessary. Just say that demographics such as age were collected. 

“Photo 1: Graphic representation of the KN-BZ and U sections. “

Please describe what the annotations on the image mean in the figure legend. 

The methodology is very difficult to read. I do not understand what was done. Maybe a more technical reviewer would understand what was done, but the clinical impact is not clear at all. 

Figures 1, 2, 3, 8, and 9 do not make sense to me and are not in English

An incredible number of tables that are confusing. I do not understand Table 19. 

Sensitivities and other results are given to 4 decimal places which again seems unnecessary

“As a result of the conducted research, a set of 23 tooth-bone indicators was identified and presented, with which it is possible to identify the position of a second premolar’s bud from digital pantomographic images, using neural modeling method. “

I really don’t know if this was demonstrated. The analysis was too complex or poorly described to be understood. 

Author Response

Dear Rewiever,

Thank you very much for your time and comments on the article. I understand that the text may seem difficult, and therefore it has been reviewed and revised once again. The tables have been moved to supplementary materials, and some of the figures have been removed.

The Bioethics Committee has determined that the study does not have the characteristics of a medical experiment, and no consent is required for the study (commission approval in the attached file).

Corrections were made in the text to the specified sentences and photos as suggested. The purpose of the study was made more precise.

As for the results given in the analysis of data using neural networks is accepted, that the results are given to 4 decimal places.

Once again, thank you very much.

Round 2

Reviewer 3 Report

Thank you for making some minor changes, but this remains very difficult to read, and still requires extensive English editing. This remains far too technical for the general readership and I cannot understand the work.